# Geographic Location, Population Dynamics, and Fruit Damage of an Invasive Citrus Mealybug: The Case of *Delottococcus aberiae* De Lotto in Eastern Spain

**DOI:** 10.3390/insects15090659

**Published:** 2024-08-30

**Authors:** Aitor Gavara, Sandra Vacas, Vicente Navarro-Llopis

**Affiliations:** Centro de Ecología Química Agrícola, Instituto Agroforestal del Mediterráneo, Universitat Politècnica de València, Camino de Vera s/n, Edificio 6C-5ª Planta, 46022 Valencia, Spain; aigavi@etsiamn.upv.es (A.G.); sanvagon@ceqa.upv.es (S.V.)

**Keywords:** detection, monitoring, sex pheromone, male captures, pest density, fruit assessment, Coccoidea

## Abstract

**Simple Summary:**

The number of invasive pests has increased worldwide, mainly due to international trade. This problem is further exacerbated by climate change and the continued restrictions on the quantity and number of pesticides imposed by international authorities. *Delottococcus aberiae*, a mealybug native to South Africa, was first detected in Europe in 2009, specifically in the Valencia Community (Spain). The identification of its sex pheromone allowed the development of a monitoring network for this pest a decade later. This network facilitated the mapping of the affected citrus area and its subsequent expansion over time. The monitoring network has revealed that from 2019 until 2023, the citrus area affected by the pest has considerably increased. This network, in conjunction with fruit damage assessments conducted at the end of the crop cycle, has also allowed us to know that the damage to the fruit is independent of the maximum annual population level of the pest. The male catches obtained during the months of March to June, when fruit setting occurs, seem to be related to economic losses due to fruit damage.

**Abstract:**

The invasive mealybug *Delottococcus aberiae* De Lotto (Hemiptera: Pseudococcidae) has rapidly spread in the Mediterranean basin since its detection in 2009 in the Valencia Community in eastern Spain. The use of sticky traps baited with its sex pheromone, (4,5,5-trimethyl-3-methylenecyclopent-1-en-1-yl)methyl acetate, has allowed to determine the geographical distribution of *D. aberiae* by means of the surveillance network described in the present work. The population monitoring of the pest over a five-year period (2019–2023) has revealed an increase from 31% to 70% of the affected citrus-growing area. The monitoring network has also allowed a better understanding of the pests’ biological cycle throughout the year. The populations start growing from March to June and reach their maximum in July–August. During autumn, there is a gradual decline in the population. Although the highest annual populations were detected in 2022 and 2023, the greatest crop losses were recorded in 2020 and 2021, with mean values near 18%. Data suggest that the damage responsible for fruit deformation, and thus the economic losses, are related to the population levels in spring (April–May) rather than those in summer (July–August). The findings of this study can be valuable for future research and development of effective pest control strategies.

## 1. Introduction

Climate change, the increase in international trade due to globalization, and the lack of effective pest control strategies—resulting from restrictions on widely used chemical substances due to their detrimental impact on the environment and human health—have increased biological invasions around the world [1,2,3,4,5]. As a result, nowadays agriculture is continuously facing invasions by new invasive arthropod pests, which are capable of causing economic losses amounting to billions of US dollars [6]. Mealybugs are among the most damaging insect pests when they appear in a new, uncolonized area without natural enemies [7]. *Delottococcus aberiae* De Lotto (Hemiptera: Pseudococcidae) is an example of these invasive mealybug pests; it is causing severe damage in the citrus-growing area in eastern Spain.

*Delottococcus aberiae* Spanish populations are native to Sub-Saharan Africa, as shown by means of molecular studies that related them to those present in Limpopo Province (South Africa) [8]. Its arrival was probably the result of international trade, as has been the case with other scale insect pests [9]. Until that moment, *D. aberiae* had never been considered a pest of economic importance and it is currently undetected in other parts of the European Union. Besides citrus, this mealybug can be found on olive trees and the roots of the flowering shrub *Chrysanthemoides monilifera* (L.) T. Norl [8] in its area of origin. *Delottococcus aberiae* is multivoltine, and its adults are sexually dimorphic, as only males are winged. Although its generations are heterogeneous and vary between years, the population that is present between May and June is the most damaging because it feeds on flower buds, producing deformed and undersized fruits [10]. During the entire growing season, both in fruit setting and developed fruits, *D. aberiae* excretes honeydew, causing the growth of sooty mold fungi *Capnodium citri* Berk (Capnodiales: Capnodiaceae) during its feeding which, in some cases, can lead to early leaf senescence and a reduction in fruit size and quality [11,12]. The first detection of *D. aberiae* in the Mediterranean basin was recorded in 2009 in Benifairó de les Valls (Valencia Community, east of Spain) [11]. Until now, there has been a continuous increase in its presence in the Spanish citrus-growing area, and there is great concern amongst authorities and farmers, due to the huge losses that can reach 80% in the worst cases [4]. The estimated economic losses from this pest exceed 110 million Euros in 2020 according to data from the Farmers Association of Valencia (AVA-ASAJA) in 2020 [13].

As in other invasive mealybug species, the control of this species was carried out by means of broad-spectrum insecticides, such as chlorpyrifos or chlorpyrifos-methyl [14]. However, the European Food Safety Authority (EFSA) concluded that concerns related to human health exist with these insecticides, particularly in relation to possible genotoxicity and developmental neurotoxicity. As a consequence, Commission Implementing Regulation (EU) 2020/18 of 10 January 2020 binds all EU Member States to withdraw all authorizations for plant protection products containing chlorpyrifos or chlorpyrifos-methyl [15,16]. Even so, the use of these products or other alternatives such as acetamiprid, paraffin oil, or spirotetramat against *D. aberiae* was not always totally effective owing to the general characteristics of mealybug pests, its waxy covering, cryptic behavior, and clumped field distribution [17]. Moreover, the short period in which the treatment must be carried out—fruit setting and the first stages of flowering—makes it difficult to affect the damaging generations. On account of the difficulty of facing this invasive pest, it was necessary to develop new control strategies based on the use of less damaging substances for the environment. In this sense, the identification and synthesis of *D. aberiae* sex pheromone, (4,5,5-trimethyl-3-methylenecyclopent-1-en-1-yl)methyl acetate, was carried out [18]. In the same study, the attraction of this new compound in both laboratory and field experiments was demonstrated. These findings allowed the development of monitoring dispensers impregnated with the synthetic sex pheromone to attract *D. aberiae* males present in infested orchards and subsequently the design of a surveillance network described in the present work. Thus, our main goals were to determine the geographical distribution of *D. aberiae* throughout the citrus-growing area of the Valencian Community and the population monitoring of the pest over a five-year period (2019–2023). Finally, a selection of eight orchards located in selected sampling units of this surveillance network was prospected in order to obtain the fruit damage at the end of each year’s crop cycle, with the aim of observing how male captures and damages at the end of the season may be related.

## 2. Materials and Methods

### 2.1. Monitoring Traps and Pheromone Dispensers

The monitoring traps used in this study consisted of 150 × 95 mm white sticky cardboards (Morelure^®^ V-Zentinel^®^, EPA SL, Carlet, Spain), baited with red rubber septa loaded with 250 µg of synthetic *D. aberiae* sex pheromone (Zentinel^®^ DAB, EPA SL, Carlet, Spain). The rubber septa were replaced every two months, as a period of efficacy ensured by the manufacturer. The traps were installed in citrus trees at the center of the orchards, tied to inner branches, at 150–190 cm above the ground.

### 2.2. Detection Network

In accordance with the provisions of local regulations (article 47 of Law 43/2002) regarding Plant Heath, the citrus-growing area of the Valencia Community was divided into grids of 5 × 5 km to establish a network of monitoring points (sampling units). The presence of potentially invasive citrus pests or the population monitoring of the existing ones is followed in each sampling unit by means of traps with their corresponding specific attractants. This official network provides the government’s phytosanitary authorities with valuable information to anticipate and know the current extent of the affected areas. In the present work, the evaluation of *D. aberiae* pest extent and its evolution was studied from 2019 to 2023, by checking male catches recorded during two months in 600 sampling units of the official detection network each year. One monitoring trap was placed in each of the 600 sampling units, collecting information from one sampling unit every 3 km^2^ of crops and covering a geographical distance of about 350 km. The minimum distance between sampling units was 1.5 km. The trap of each sampling unit was always located at the center of an orchard.

Trap catches were recorded from mid-May to mid-June (30-day sampling) and from mid-June to mid-July—the months expected to suffer the higher populations [16]. Male capture values (per trap and month) recorded in each sampling unit were represented on maps by means of a scale with circles of different colors and sizes. The maps were created with ArcMap 10.5 software (Esri, Redlands, CA, USA).

### 2.3. Monitoring Network

In addition to studying the population dynamics of the pest, a complementary monitoring network was established to follow populations monthly throughout the year. For this purpose, part of the 600 sampling units of the detection network were used to obtain the pest pressure and population dynamics. Monthly male catches were registered in the 58 sampling units, where captures were over 0.1 males/trap·day (MTD) during 2019 (from May to November 2019). In the case that two neighboring traps of the grid were over 0.1 MTD, only the trap with the highest number of catches was selected. In this way, the points in areas with very high populations were reduced by half, thus avoiding their overrepresentation on the maps. After the first year, the number of sampling units increased to 78 for the rest of the study (from March 2020 to December 2022) due to the increasing number of catches in new locations. Monthly captures obtained in the 58 or 78 sampling units were averaged and plotted.

### 2.4. Fruit Damage Caused by D. aberiae in Damage Assessment Plots

In order to know how the population level could influence fruit damage, 8 of the 78 orchards used in the monitoring network were assessed from 2020 to 2023. The assessment was based on two parameters: (1) the fruit damage at the end of the crop season each year and (2) the study of male flight during the months when the pest population’s growth coincides with fruit setting. For the fruit damage assessment, 20 trees were selected in the center of each orchard, with 5 fruits per orientation (N, E, W, S) visually inspected (a total of 400 fruits/orchard). Fruit damage was categorized according to Figure 1. Fruits in categories 0 and 1 were considered marketable, while those in categories 2 and 3 were considered without commercial value. The study of the male flight was carried out according to the male captures recorded in the monitoring traps of each orchard during March, April, May, and June (2020–2023).

### 2.5. Statistical Analysis

The number of males per trap and day counted in the monitoring network from January to December was log-transformed (ln[captures + 1]) to normalize residual data distribution and homogenize the variance. A one-way ANOVA (Fisher least significant difference [LSD] test at *p* < 0.05) was performed to check for the statistical differences between the values for each month over the years.

With data from damage assessment plots, three different analyses were performed. The percentage of fruit damage, with the different categories described (0—healthy fruit; 1—slight fruit deformation; 2—fruit deformation with a clear loss of asymmetry; and 3—aberrant fruits), was individually studied to check for statistical differences among years via a one-way ANOVA (Fisher least significant difference [LSD] test at *p* < 0.05). In this case, the data fulfilled the homoscedasticity and the normal distribution requirements to be analyzed without data transformation. Then, percentages of unmarketable fruits (grouping damage categories 2 and 3) were compared among years in the same way, via a one-way ANOVA (Fisher least significant difference [LSD] test at *p* < 0.05) and without data transformation. In the case of the monthly flight of males from March to June, the data was log-transformed (ln[captures + 1]) to normalize residual data distribution and to homogenize the variance. A one-way ANOVA (Fisher least significant difference [LSD] test at *p* < 0.05) was performed to check for the statistical differences between the values for each month over the years.

## 3. Results

### 3.1. Detection Network

The capture data obtained in the first sampling, mid-May to mid-June 2019, showed that the pest was already present in 32.18% of the citrus-growing area of the region ten years after its introduction, with varying degrees of pressure (Table 1). The highest captures were found at the nearest locations to the original pest introduction in 2009 (black dot in Figure 2A), South-Castellón (50 km) to North-Valencia (8 km), whereas the captures decreased up to fewer than one male per trap and day (MTD) in the most distant locations, North-Castellón (120 km) and South-Alicante (250 km). The number of plots with positive detections of *D. aberiae* was slightly lower in 2020 than in 2019, with 144 and 196, respectively, which represents a percentage of the area infested by the pest of approximately 32% in 2019 and 2020. However, the male population catches were much higher in 2020 than in the previous year, as evidenced by the higher number of red dots (>20 MTD) both in the area of introduction and in other areas located to the north of the introduction area (Figure 2A,B), increasing the citrus-growing area with catches of more than 20 MTD from 0.16% to 13.30% (Table 1).

In the following years, there was a general increase in the number of detections until 2022, with 284 and 354 detections in 2021 and 2022, respectively (Figure 2C,D). The total citrus-growing area infested with this pest increased from 31.53% in 2020 to 48.44% in 2021 and 70.94% in 2022 (Table 1).

In comparison to the previous year, the number of detections in 2023 remained stable, with only eight new detections (Figure 2E) and a slightly lower infested area, at 66.17%, as well as very similar percentages of pest pressure in the infested areas (Table 1). Accordingly, pest populations consolidated and increased both to the south and to the north at locations distant from the original point of introduction (60–180 km). Interestingly, male catches were reduced at the initial pest hotspots, as evidenced by the lowest density of red spots in this area on the 2021 and 2022 maps (Figure 2C,D), with a stabilization in 2023 (Figure 2E). The complete series of maps displaying the monthly male trapping levels can be found in Appendix A.

### 3.2. Monitoring Network

Data collected once a month from the traps of the monitoring network (58–78 traps) highlighted the varying dynamics of *D. aberiae* populations in the different years of study (Figure 3).

The period with the lowest catches was consistent throughout the study, which was February (Figure 3). However, we can find great differences in monthly male catches throughout the years. In 2019, male catches did not start to rise significantly until May, and two flight peaks were identified, in July and October, with similar mean numbers of captures (ca. 33 MTD) (Figure 3).

From March to June, the period when severe fruit damage occurs, an increasing number of captures were observed, with a mean of fewer than 4 MTD in March, and captures in some years exceeded 60 MTD in June (Table 2). In 2020, male catches began to increase earlier, exceeding 6 MTD in April (Table 2) and reaching the maximum flight peaks during June–July (Figure 3). After that, captures began to decrease, but a second not very prominent peak was also detected during October, similar to that of 2019. In 2021, the captures in March were significantly lower than in the other years of the study, but captures also began to increase in April with values comparable to 2020 (Table 2). The maximum flight peak was reached during August, almost 2 months later than in 2020, but at similar levels (ca. 60 MTD). Interestingly, the second peak usually observed during fall was virtually unnoticed during 2021. The population build up did not take place in 2022 until May. In fact, April captures were the lowest compared to the other four years (Table 2). This year the maximum flight peak coincided with that of 2021, during August, but with the highest captures recorded in the study, three times higher pest pressure (59 vs. 187 MTD in 2021 and 2022, respectively). Subsequently, the fall peak was detected in November 2022, once again one month later than in 2019 and 2020. Finally, population dynamics in 2023 were similar to those observed in 2022, with the population starting to rise in May to reach maximum male captures in August, but at the same levels as in 2021.

### 3.3. Fruit Damage Caused by D. aberiae

Fruit damage data recorded during 2020 and 2021 in the eight selected orchards were statistically similar, with an average of 17.9% and 18.7% of unmarketable fruit (Table 3). However, a significant reduction in fruit damage was observed in the two following years, dropping to 3.0% in 2022 and 0.5% in 2023 (*p* < 0.05), even though 2022 had the highest male population peak during the study period (Figure 2). These data refer to the mean percentages of fruit with damage categories 2 and 3, considered unmarketable. If we consider the percentage of fruit with damage level 1 (marketable but with slight deformations) (Table 3), it was significantly higher in 2020 (32.5% of the fruit) and moderate in 2021 (13.2%). In 2022, it was very low, showing that although the population was very high in the summer season of 2022, fruit damage was almost negligible. In 2023, the results were similar to those of the previous year.

The mean number of monthly male captures in the eight selected orchards (Table 4) revealed initial low values in March, which increased by June.

In March, all the values were below 2.33 MTD, and there were no significant differences among the four years of study (*p* > 0.05). Catches increased in April but with significant differences among the different years. In May, the values reached in 2020 and 2021 were notably the highest of the four years (*p* < 0.05). In all cases, male captures increased in June, with values near 100 MTD for 2020, 2021, and 2023, whilst in 2022, the number of captures remained ten times lower (*p* < 0.05).

## 4. Discussion

Spain is the largest citrus producer in the European Union (60% of the total production) and the fifth largest in the world, with an annual production of nearly 6 million tons. Spain is also the main global fresh citrus exporter, with 4 million tons of citrus exported annually [19]. In the whole of Spain, the Valencia Community generates 60% of the national citrus production, with more than 155,000 ha cultivated and nearly 3 million tons of fruit production [20]. Given the importance of the crop and the losses already caused by this mealybug, the development of effective and sustainable specific control methods is crucial. 

The morphology, cryptic behavior, and clumped distribution of mealybugs render them difficult to detect during winter or in the initial annual generations at the onset of winter [7]. *Delottococcus aberiae* is not an exception, and the use of traditional methods to monitor its presence, such as visual inspections searching for individuals or sooty molds of honeydew residuals, can be ineffective during the early stages of the invasion. The efficacy of these methods increases when population densities are higher, but this is when losses have already occurred [7,21,22]. Currently, based on behavioral and physiological knowledge of mealybugs, the availability of some of their sex pheromones has allowed the development and implementation of pheromone-baited traps, which have the advantage of avoiding the need for tedious and difficult manual sampling methods while increasing the sensitivity of the monitoring activity [7,23,24]. Moreover, the use of sex pheromones allows for the identification of pests without the need for extensive taxonomic expertise in mealybugs. This is especially important given that in our Mediterranean citrus crops, different species of mealybugs can be found sharing the same niche: the citrus mealybug (*Planococcus citri* Risso), the long-tailed mealybug (*Pseudococcus longispinus* (Targioni Tozzetti)), the obscure mealybug (*Pseudococcus viburni* Signoret), and *D. aberiae*.

Just as was the case for other mealybug species [25,26], the identification and synthesis of *D. aberiae* sex pheromone [18] have enabled the detection of its spread throughout crop production areas. The development of the present surveillance network has provided a large amount of information regarding pest presence and its pressure throughout the citrus-growing area over the five-year study period. The evolution of the locations where the pest has been present from 2019 to 2023 shows an increase in the total area affected by *D. aberiae*, especially at distant locations from the original area of introduction. There are several zones in which, despite the presence of the pest, significant economic losses have not been registered since in the first recordings there were fewer than 1 MTD. However, in plots with higher male catches, over 28 MTD in the sum of April and May population catches, fruit damage has consistently increased. Accordingly, significant correlations between male captures and fruit damage may be found, although it is necessary to have more data series from more locations to be able to establish firm conclusions.

Our results showed that annual fruit damage was not directly related to the maximum annual male catches. The maximum population level was recorded in 2022 (Figure 3), when fruit damage was significantly lower compared to 2020 and 2021 (Table 3). However, looking at the male captures recorded in April and June in the orchards where damage was assessed (Table 4), there were significantly higher values in 2020 and 2021 than in 2022, while April is the main month of citrus flowering in Spain and therefore of fruit setting [10]. Previous works have established citrus phenological stages between flowering (BBCH 69–71, March–April in eastern Spain conditions) and fruits with a diameter of 25–30 mm (BBCH 71–74, around July in eastern Spain conditions) as the most susceptible period for *D. aberiae* damage [10,14]. This might justify why, despite the highest captures, damage was low during 2022, as summer populations are not responsible for those damages. Some authors suggested that the highest percentage of damage is recorded when *D. aberiae* attacks the initial stages of fruit development and that no damage was observed when *D. aberiae* was in contact with fruits exceeding 30 mm in diameter [10]; moreover, fruits obtained from infested flowers in March and April also appeared distorted. Therefore, the main fruit damage inflicted by *D. aberiae* could be related to male catches in April that mate and provoke offspring, leading to high populations in May and June. Therefore, a better understanding of the relationship between male flights and the presence and composition of the preimaginal population, as well as the determination of a level of catches that trigger fruit damage during these months, might be crucial to suggest insecticide treatment timings to the farmers, especially in the current situation of restrictions on the use of pesticides.

It is well known that predicting future pest damage using monitoring tools is a key point for the implementation of IPM. Monitoring tools can be based on early field counts or insect captures in baited traps, with the former being more time- and work-consuming but essential when no pheromone or attractants are available. Trap counts are especially valuable when a powerful specific pheromone is available and for cryptic species that can easily pass unnoticed. In the case of *D. aberiae*, both circumstances occur: a very effective sex pheromone is available, and the insect is very cryptic. Therefore, the use of monitoring traps can be a great tool to determine treatment thresholds. Previous research has been carried out to determine the treatment threshold in *D. aberiae* [4], but until now, no research has been carried out using pheromone monitoring traps to determine these thresholds. In this work, high male populations were detected during autumn. However, these high male catches do not correspond to the female presence in the aerial part of the tree, as described by Martínez-Blay et al. [27]. This difference may be due to the fact that the females take refuge in the roots in autumn–winter and ascend to the aerial part in spring–summer. Pheromone-baited traps are capable of detecting males that emerge from below the ground and therefore detect populations that aerial surveys are not capable of detecting [28].

The time series of the area occupied by *D. aberiae* over the years demonstrates a significant increase in the extent of the infestation, with 30% of the area occupied in 2019 increasing to almost 70% in 2023. This rapid expansion is indicative of the high dispersal capacity of this species. The primary means of transporting pests over long distances is the movement of plant material through the horticultural and ornamental trade [29]. The ease of this kind of dispersal for mealybugs is attributable to their size and their capacity to go unnoticed hidden in nooks of the plant material, such as fruit protuberances and leaf axils. It can be reasonably assumed that the spread of the pest during farming labors is likely to occur, especially during harvest when workers and harvesting and transport equipment, mainly fruit crates, travel from one orchard to another. This may explain the presence of the pest in such distant locations in a relatively short period of time, with distances of over 200 km being covered in a few years given that the pest has also been detected in the neighboring regions of Catalonia and Murcia [30]. Fruit traders collect the product, also transporting the pest. However, in the rapid expansion of mealybugs as pests, short-distance dispersal should not be underestimated, as it may also play a role in the first outbreaks in areas in close proximity to the original detection site. Wind is the main spreading agent for crawlers in short distances, together with ants [31,32], which attend mealybug colonies for their honeydew. Indeed, there are studies in which mealybug control improves with the exclusion of ants [33,34]. Another factor that plays a significant role in the rapid spread of *D. aberiae* is the lack of natural enemies in our region. However, a parasitoid complex that is capable of controlling the pest naturally exists in the native area of the pest [35].

A comparison of population levels and dynamics over time reveals that high populations in one year do not necessarily mean a higher level in the subsequent year. This is evidenced by the data from the final years of this study, 2021 versus 2022 and 2023. Martínez-Blay et al. [27] conducted a three-year study (2014, 2015, and 2016), examining the population dynamics of all *D. aberiae* stages. Given the unavailability of the synthetic pheromone, they employed sticky traps baited with virgin females to monitor *D. aberiae* male flight and detected two main peaks, one occurring in spring and the other in summer (April and mid-June, respectively). Interestingly, these peaks differed slightly between years, similar to our observations. They identified (1) climate conditions and (2) the quality of the feeding substrate as possible causes of these variations. With regard to climate, lower temperatures in the early spring could lead to a delay in the population development and, consequently, a delay in the male captures. In summer, in the East of Spain, it is common to experience west winds that cause high temperatures and very low relative humidity (Foehn winds), meteorological episodes that can negatively affect mealybug first instars and reduce pest populations [27]. Moreover, rain events may diminish male flights and their abundance, as described in 2015 in that study. With regard to the feeding substrate, it is known that mealybugs tend to hide in the shoots during the overwintering period, and their development is in alignment with the growth of the shoots. After this period, young stages set themselves in the flower buds, the calix of young fruits, and finally on fruits, where individuals find higher-quality food, leading to higher fecundity and population outbreaks [27,36].

In our five-year study, a population decrease was observed in April 2022 compared to the same month of the previous year. This reduction in male catches may be linked to specific meteorological events that occurred in April 2022. During that month, there were two days when the minimum temperatures fell below 2 °C, a figure that typically does not fall below 5 °C. Moreover, there was an unusual increase in rainfall, with a monthly average exceeding 80 L/m^2^, compared to the typical values that range between 20 and 50 L/m^2^. Nevertheless, further investigation with more study years would be needed to establish a significant correlation between climate conditions and *D. aberiae* populations. This information underscores the necessity for a comprehensive phenology model that can elucidate the proportion of the different developmental stages and their abundance throughout the year.

## 5. Conclusions

This work has demonstrated the utility of pheromones as a tool for monitoring and detecting new invasive pests such as *D. aberiae*, as well as the need to identify new insect pheromones. It was observed that, ten years after the first detection of this mealybug in 2019, its populations have spread throughout most of the citrus-growing region, highlighting the rapid and easy spread of this type of pest. These results highlight the necessity of the implementation of precautions in the future, particularly regarding the management of plant material and tools, specifically during harvest. The data obtained from this surveillance network indicate that fruit deformation, the damage that causes the highest economic losses, does not occur in the months with the highest population density. Rather, it occurs in early spring, coinciding with fruit setting. This information may be of great interest in the future for establishing treatment thresholds. The importance of developing new tools for early identification or detection of invasive pests in order to anticipate potential threats is also reflected in this work.

## Figures and Tables

**Figure 1 insects-15-00659-f001:**
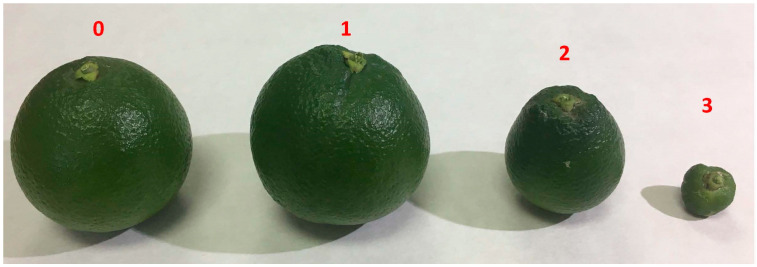
Categorization of fruit damage in four levels: (0) healthy fruit; (1) slight fruit deformation around the calyx; (2) fruit deformation with a clear loss of asymmetry and/or size reduction; (3) aberrant, dwarf, and totally deformed fruits.

**Figure 2 insects-15-00659-f002:**
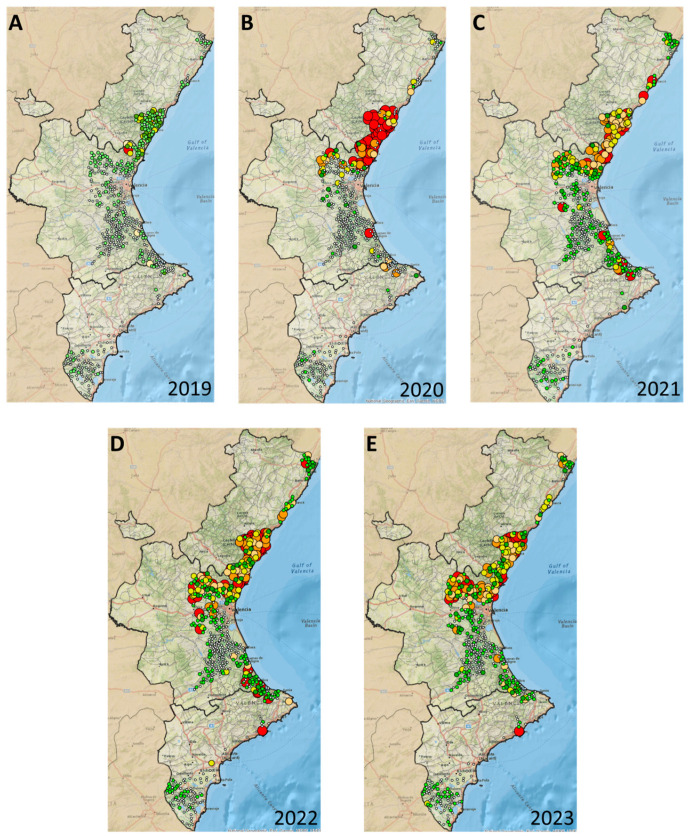
*Delottococcus aberiae* male catches recorded at each of the 600 sampling units of the detection network from mid-May to mid-June in each study year: (**A**) 2019, (**B**) 2020, (**C**) 2021, (**D**) 2022, and (**E**) 2023. Captures were obtained in pheromone-baited monitoring sticky traps and were represented according to the following scale (males/trap/day): 0 (white dot), 0–1 (light green dot), 1–3 (dark green dot), 3–6 (yellow dot), 6–10 (light orange dot), 10–20 (orange dot), and >20 (red dot). The black mark in 1A indicates the original pest introduction.

**Figure 3 insects-15-00659-f003:**
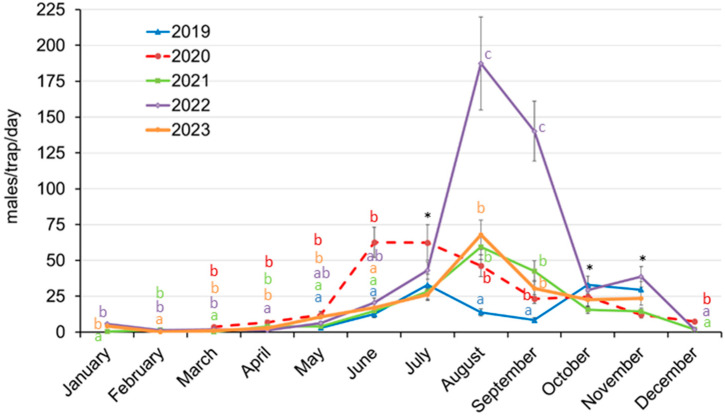
Daily mean (±standard error) male catches recorded on the 58 or 78 traps of the monitoring network each year throughout the study period. Different letters indicate significant differences among years for the same month, and the asterisks indicate the absence of statistical differences by ANOVA followed by the Fisher LSD test (*p* < 0.05) (more details in Appendix A).

**Table 1 insects-15-00659-t001:** Percentage (%) of the total Valencian citrus-growing area infested with *D. aberiae* and of the area affected by different male *D. aberiae* trapping levels (0, 0–3, 3–10, 10–20, and >20 males per trap and day) in the traps counted in mid-May and mid-June from 2019 to 2023.

Year	Total Area	0	0–3	3–10	10–20	>20
2019	32.18	67.82	28.24	3.12	0.33	0.16
2020	31.53	68.47	6.57	4.93	3.94	13.30
2021	48.44	51.56	26.11	7.39	5.42	9.36
2022	70.94	29.06	32.18	11.49	5.42	21.51
2023	66.17	33.83	30.38	11.99	7.22	16.42

**Table 2 insects-15-00659-t002:** Daily mean (±standard error) male catches recorded on the 58 or 78 traps of the monitoring network in the months of March, April, May, and June each year throughout the study period. Different letters indicate significant differences between years, ANOVA followed by the Fisher LSD test (*p* < 0.05).

Year	Males/Trap/Day			
	March ^1^	April ^2^	May ^3^	June ^4^
2020	3.69 ± 0.81 b	6.73 ± 1.56 b	12.19 ± 2.30 b	62.67 ± 10.64 b
2021	0.10 ± 0.02 a	4.03 ± 0.81 b	3.87 ± 0.68 a	14.79 ± 3.25 a
2022	2.03 ± 0.50 b	1.41 ± 0.25 a	6.39 ± 1.15 ab	20.69 ± 3.25 ab
2023	1.09 ± 0.19 b	2.91 ± 0.43 b	10.65 ± 1.49 b	17.16 ± 3.34 a

^1^ F = 21.94; df = 212.3; *p* < 0.0001. ^2^ F = 3.01; df = 208.3; *p* < 0.05. ^3^ F = 4.12; df = 214.3; *p* < 0.05. ^4^ F = 2.72; df = 2202.3; *p* < 0.05.

**Table 3 insects-15-00659-t003:** Results of the percentage of fruits (mean ± standard error) for each level (unmarketable with damage levels 2 and 3) throughout the four years of fruit assessment. Different letters indicate significant differences between years, ANOVA followed by the Fisher LSD test (*p* < 0.05).

Year	Fruit Damage Level (%)				
	0 ^1^	1 ^2^	2 ^3^	3 ^4^	Unmarketable ^5^
2020	49.6 ± 10.1 a	32.5 ± 5.9 a	8.8 ± 2.5 a	9.1 ± 3.1 a	17.9 ± 4.8 a
2021	68.1 ± 9.2 a	13.2 ± 6.5 b	7.6 ± 2.2 a	11.1 ± 3.1 a	18.7 ± 4.8 a
2022	94.8 ± 1.1 b	2.2 ± 0.6 b	1.7 ± 0.4 b	1.3 ± 0.3 b	3.0 ± 0.7 b
2023	98.3 ± 0.4 b	1.2 ± 0.2 b	0.3 ± 0.1 b	0.2 ± 0.1 b	0.5 ± 0.2 b

^1^ Unaffected fruits (F = 9.39; df = 24.3; *p* < 0.005). ^2^ Slightly deformed fruits (F = 10.06; df = 24.3; *p* < 0.005). ^3^ Deformed and size-reduced fruits without symmetry (F = 4.78; df = 24.3; *p* < 0.05). ^4^ Aberrant fruits (F = 4.26; df = 24.3; *p* < 0.05). ^5^ Unmarketable fruits (F = 4.86; df = 3.24; *p* < 0.05).

**Table 4 insects-15-00659-t004:** Catches per trap and day (mean ± standard error) recorded on the eight damage assessment plots during each month throughout the first months of flight activity (March, April, May, and June) for the four years of fruit assessment. For each month, different letters indicate significant differences between years, ANOVA followed by the Fisher LSD test (*p* < 0.05).

Year	Males/Trap/Day			
	March ^1^	April ^2^	May ^3^	June ^4^
2020	-	3.37 ± 0.57 b	60.54 ± 14.08 a	84.38 ± 17.15 a
2021	1.44 ± 0.44	10.22 ± 1.97 a	18.40 ± 3.51 b	82.67 ± 20.33 a
2022	2.09 ± 1.97	1.19 ± 0.57 c	7.92 ± 3.84 c	12.80 ± 3.15 b
2023	2.33 ± 0.85	6.45 ± 1.12 ab	13.27 ± 3.98 bc	91.83 ± 15.49 a

^1^ F = 0.94; df = 2.17; *p* = 0.41. ^2^ F = 12.21; df = 3.23; *p* < 0.005. ^3^ F = 11.12; df = 3.23; *p* < 0.005. ^4^ F = 9.12; df= 3.24; *p* < 0.005.

## Data Availability

All data supporting the findings of this study are available from the corresponding authors upon reasonable request, subject to the restrictions imposed by the project’s funding agency.

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
