# Peer review of "Geographic Location, Population Dynamics, and Fruit Damage of an Invasive Citrus Mealybug: The Case of Delottococcus aberiae De Lotto in Eastern Spain"

_insects, 2024, doi:10.3390/insects15090659_

Round 1

Reviewer 1 Report

Comments and Suggestions for Authors

The paper deals with an interesting topic, even if for the target species, of local interest for Spain at the moment,  but the approach on the use of pheromone traps is certainly broader.

A snapshot is made of what is the situation of the diffusion of the target species in the monitored area of Spain. The data reported, even if in a very simple way, with basic statistics, are interesting, while the statements made in the conclusions are overestimated in various points. I suggest a major revision of the manuscript, to rearrange all the material and methods and the conclusion. A change of perspective is needed. In my opinion, we cannot talk about the distribution of a species without referring to a model connected to dedicated algorithms.

Comments on the Quality of English Language

Even though I am not a mother tongue, I think English needs to be thoroughly revised.

Author Response

The paper deals with an interesting topic, even if for the target species, of local interest for Spain at the moment, but the approach on the use of pheromone traps is certainly broader. A snapshot is made of what is the situation of the diffusion of the target species in the monitored area of Spain. The data reported, even if in a very simple way, with basic statistics, are interesting, while the statements made in the conclusions are overestimated in various points. I suggest a major revision of the manuscript, to rearrange all the material and methods and the conclusion. A change of perspective is needed. In my opinion, we cannot talk about the distribution of a species without referring to a model connected to dedicated algorithms.

We are grateful for your suggestions, which have been taken into account in the revision of the article. Although the study is based on a local perspective, it is important to note that the pest in question has the potential to emerge in other countries within the Mediterranean Basin at any given time. We totally agree that our study depicts the situation of the pest, but our objective was never to create a predictive model of the pest's behaviour. Accordingly, we have tried to lighten our statements, as the reviewer suggests. The study was focused on demonstrating the evolution of the pest through the use of a monitoring network, with the aim of obtaining as much information as possible that can be used in the future.

Title

It does not seem completely pertinent and is a little misleading: studies on distribution of a pest should provide models or dedicated algorithms applicable to the collected data. What is done in the course of the work is a picture of where the target insect has spread in the monitored area. So I suggest to rearrange the title maintaining the term detection and population dynamics and delete the distribution term.

Title has been changed following the suggestions. Pest distribution changed to Geographic location (L2).

Introduction

line 57: Where the preimaginal population of the mealy bugs eat in the other period? They don’t' live on fruits? If yes, why they do not cause damage by feeding on fruits? Maybe the damage has a less extent? Please clarify

Generations are overlapped, so all instars can be found feeding on plant parts during the whole growing season. Information added and clarified on L60-64.

L59-61: During the entire growing season, both in fruit setting and developed fruits, D. aberiae excretes honeydew causing the growth of sooty mold fungi Capnodium citri Berk (Capnodiales: Capnodiaceae) during its feeding [10].

Changed as suggested (L63)

Line 77: reference 14; it does not seem that the citation is correct: the paper does not contain indication about Delottococcus control

This reference is correct. It refers to general characteristics of mealybugs such as D. aberiae.

Line 91: …Valencia community. Then the other part of the phrase on the dimension of the grid should be move in material and methods chapter

Done. Moved to material and methods section.

Line 95: how many traps per units? Please, clarify. Also this part should be move in material and methods part.

One trap per sampling unit. Moved to material and methods section.

Line 98: 79 (?) or 58 and 78 as describes in material and methods?

Thanks, 78.

Material and methods

2.1

Line 109: which is the distance among the traps in the fields? How many traps per hectare? Where traps were positioned into the fields? (at the border, inside...?)

Each sampling unit comprised one trap, and these were positioned at the center of an orchard. Information added (L.107-119).

2.2 Title: Delete distribution and leave detection as in results chapter. It is not very clear how the monitoring of only two months could be useful for detection and distribution. For an early detection the survey period should be longer than that and for distribution how this can help us? Maybe you can change the title of this paragraph. Delottococcus here not in italics, being Italics the rest of the period.

The objective of this network was not to provide an early detection of the pest; rather, it was to obtain a representative picture of its presence along our geography throughout the years. Therefore, the months in which the pest is theoretically more abundant were selected as the optimal time frame for data collection. If we are not able to detect a male during the months when populations are maximum, the probability of detecting the pest during periods with low population levels should be lower. It is intuitively obvious that the probability of catching a male when population is abundant is higher than when populations are low. In addition, the title has been changed as suggested (L. 99).

Line 115: Here you could insert the part written in the introduction chapter, describing in detail the organization of the monitoring network.

The description of the monitoring network has been finally inserted in L125.

Line 116: 350 km; is this a linear measure? which is the covered areas It is not clear enough

It is a geographical distance (L. 118.). The area covered by the network is of about 1800 km2.

Line 117: It is not clear how the traps are arranged in the monitoring area. Then the total number of the traps is 3*600= 1800??

The total number is 600; one trap per sampling unit (included in the description, L116).

Line 121: Figure 1 proposes images of fruits, the reference should be to S1 figure maybe. however, also fig S1 in the appendix is not clear enough, it is too small, we can’t' appreciate the details

It was a typo; the reference to the figure has been deleted from this section. Thanks.

2.3

Line 128: which is the minimum distance between two traps?

3 Km, added in L118

Line 140: insert in a bracket (N,E,O,S)

Thanks; inserted in L.145

Line 163: This seems the same analysis written in the first period, which is the difference?

One analysis was performed for each damage category, and the other for the unmarkatables (categories 2 and 3 together). Information added in L.161-174

Results

3.1 Title: The title of these chapters for results are not the same of material and methods; this one is better, change in material and methods Line 177-179: not very clear

Done.

Figure 2: we can't appreciate the details in the figure

To have a bigger figure, we have changed the design but we are not sure if this will be acceptable for the journal.

Figure S1: The figure is very difficult to read for the tiny dimension of dots

The size of the Figure S1 has been increased. You can zoom in on it without losing resolution.

Line 329: Yes, I agree. This should be an hypothesis but need a deeper analysis of the data and maybe the study of a model for the distribution of the species.

Yes, this was just a hypothesis, and we are aware of the scope of this study and that more data and modeling are needed to draw powerful conclusions.

Line 347: This is an important assessment, but maybe perhaps further studies are needed to better understand the relationship between male flights and the presence and composition of the preimaginal population to better time treatments

Clarification added. L346-350.

Line 361: The reference n.26 is not Martinez-Blay et al ,but González-Gaona et al.

Thanks, corrected L.

Line 412: I agree: many of your final consideration are partial without this kind of study.

Yes, it is necessary a deep knowledge of the biology of this pest and their development in our agroecosystem.

References

Line 468: this reference is inserted twice, see ref 4

Thanks.

Reviewer 2 Report

Comments and Suggestions for Authors

Comments for authors

Summary

Line 11: continuous restriction or application of numerous pesticides? Please recheck and explain.

Abstract

Line 32-33: Although the highest annual populations were detected. Please add word annual for better understanding.

The abstract section lacks a conclusive statement/recommendation that how this study can provide useful insights regarding its effective management keeping in view its biological cycle and maximum infestation period. This must be added at the end of abstract.

Introduction

Line 42: restriction of the most widespread active substances for pest control. Please explain this fact for better understanding.

Line 44: facing with new invasive arthropod pests. Better to write as “facing invasions of new arthropod pests,

Line 44: billions “billions what USD or some other currency”? Better to write as “which are able to cause economic losses of billions of US $”

Line 58-59: its feeding on. Better to write “it feeds on”

Line 61: sooty mold fungi Capnodium citri. Please also mention how this fungus affects crop growth and yield.

Line 70-74: This information must be escorted with some latest citations.

Materials and Methods

Section 2.1: The rubber septa were replaced every two months, as a period of efficacy ensured by the manufacturer. What happens in case of abrupt ecological conditions such as rainfall? Did you replace septa after rainfall? As, the distance is 350 Km, what happened regarding this septa replacement, if it rains in some parts of studied area?

Line 129-130: In this way, the points in areas where the population is very high are reduced by half and thus, we avoid. It should be “was very high” “were reduced” and “we avoided”. The given information should be past tense.

Line 143: Is there any reference for this fruit categorization?

Results

Line 172: 32.18%. Please be consistent with the values mentioned in table. Same implies with rest of the results.

Line 210: replace highlight with highlighted

Figure 3: Better add small alphabets highlighting significance level as mentioned in Table 2-4.

Describe full form of SE in figure caption and table headings or footnotes.

Line 227: Replace can be with were. Please be careful with grammatical errors.

Line 248: No need to mention (F=4.86, df=3,24, P<0.05) in results description. However, only P value will be enough if you want to write it.

Line 283: replace obtain with obtained

Line 285: [16] is this a citation? Why are you putting a citation in the result section? The logical reasoning must be explained in discussion section.

Discussion

This section has scattered information. I suggest authors should stick to their results and compare them with previous findings to support current results. Moreover, logical reasoning pertaining to current outcomes must also be elaborated in detail. The information in first two paragraphs can be summarized.

Author Response

We are grateful for your suggestions, which have been taken into account in the revision of the article. We have enriched the discussion with our study data and changed deeply the conclusions section.

Summary

Line 11: continuous restriction or application of numerous pesticides? Please recheck and explain.

This has been rephrased (L11-13). We meant continuous restrictions in the use of pesticides (quantities and number of active substances).

Abstract

Line 32-33: Although the highest annual populations were detected. Please add word annual for better understanding.

Added “annual” as suggested (L31.)

The abstract section lacks a conclusive statement/recommendation that how this study can provide useful insights regarding its effective management keeping in view its biological cycle and maximum infestation period. This must be added at the end of abstract.

Thanks. Modified in L32-35.

Introduction

Line 42: restriction of the most widespread active substances for pest control. Please explain this fact for better understanding.

Rephrased for clarity (L41-43).

Line 44: facing with new invasive arthropod pests. Better to write as “facing invasions of new arthropod pests,

Done (L44-45).

Line 44: billions “billions what USD or some other currency”? Better to write as “which are able to cause economic losses of billions of US $”

Done (L45)..

Line 58-59: its feeding on. Better to write “it feeds on”

Done (L59-60).

Line 61: sooty mold fungi Capnodium citri. Please also mention how this fungus affects crop growth and yield.

Included in L62-64.

Line 70-74: This information must be escorted with some latest citations.

Citations added. L64

Materials and Methods

Section 2.1: The rubber septa were replaced every two months, as a period of efficacy ensured by the manufacturer. What happens in case of abrupt ecological conditions such as rainfall? Did you replace septa after rainfall? As, the distance is 350 Km, what happened regarding this septa replacement, if it rains in some parts of studied area?

The rubber is impregnated by immersion, the pheromone is adsorbed in the material and does not lose significant effectiveness after rainfalls because the adsorbed pheromone is not completely washed. Rather than septa, the traps themselves are more susceptible to heavy rain episodes due to wind and water. the cardboard sticky trap can fall or break in heavy rain episodes. When this happened, this point was discarded from the study during that month.

Line 129-130: In this way, the points in areas where the population is very high are reduced by half and thus, we avoid. It should be “was very high” “were reduced” and “we avoided”. The given information should be past tense.

Done L. 132-134

Line 143: Is there any reference for this fruit categorization?

This fruit categorization is not included in any scientific publication. This categorization is being used in technical reports to facilitate a better understanding of the type of damage being referred to.

Results

Line 172: 32.18%. Please be consistent with the values mentioned in table. Same implies with rest of the results.

Thanks; changed as suggested.

Line 210: replace highlight with highlighted

Done.

Figure 3: Better add small alphabets highlighting significance level as mentioned in Table 2-4.

Done, small letters have been included in the Figure as suggested. All the statistics have been included in a supplementary table.

Describe full form of SE in figure caption and table headings or footnotes.

Done.

Line 227: Replace can be with were. Please be careful with grammatical errors.

Done.

Line 248: No need to mention (F=4.86, df=3,24, P<0.05) in results description. However, only P value will be enough if you want to write it.

Done.

Line 283: replace obtain with obtained

Done.

Line 285: [16] is this a citation? Why are you putting a citation in the result section? The logical reasoning must be explained in discussion section.

We totally agree. This paragraph has been included in the discussion section. L329-338

Discussion

This section has scattered information. I suggest authors should stick to their results and compare them with previous findings to support current results. Moreover, logical reasoning pertaining to current outcomes must also be elaborated in detail. The information in first two paragraphs can be summarized.

Thanks for this comment. We tried to include data of our study to increase the logical sequence of the text and tried to summarize the two first paragraphs of the text.

Round 2

Reviewer 1 Report

Comments and Suggestions for Authors

I thank the Authors for having accepted my suggestions. In this new form it seems that your manuscript is more in line with what you wanted to illustrate. Surely the monitoring of newly introduced insects is absolutely fundamental to verify their diffusion in the areas of interest and the use of specific pheromones is of great utility.

Comments on the Quality of English Language

I noticed just a few typos at line 341; 442; 447; 465. I suggest a careful rereading